# A Kernel Density Estimation based Quality Metric for Quality Assessment of Obstetric Ultrasound Video

**Jong Kwon**[1], **Jianbo Jiao**[1,2], **Alice Self**[3], **J. Alison Noble**[1], **Aris Papageorghiou**[3]
[1]Department of Engineering Science, University of Oxford
[2]School of Computer Science, University of Birmingham
[3]Nuffield Department of Women's & Reproductive Health, University of Oxford
`jong.kwon@keble.ox.ac.uk`

## Abstract

Simplified ultrasound scanning protocols (sweeps) have been developed to reduce the high skill required to perform a regular obstetric ultrasound examination. However, without automated quality assessment of the video, the utility of such protocols in clinical practice is limited. An automated quality assessment algorithm is proposed that applies an object detector to detect fetal anatomies within ultrasound videos. Kernel density estimation is applied to the bounding box annotations to estimate a probability density function of certain bounding box properties such as the spatial and temporal position during the sweeps. This allows quantifying how well the spatio-temporal position of anatomies in a sweep agrees with previously seen data as a quality metric. The new quality metric is compared to other metrics of quality such as the confidence of the object detector model. The source code is available at: https://github.com/kwon-j/KDE-UltrasoundQA

## 1 Introduction

Obstetric ultrasound scanning is a vital part of antenatal care, as it allows us to accurately date a pregnancy, identify pregnancy risk factors, check the health of the fetus and much more (Crino et al., 2013; Salomon et al., 2011; 2019). Although ultrasound is considered a cheap and often portable imaging modality, there is still a shortage of availability of obstetric ultrasound scans in limited resource settings (Ngoya et al., 2016; Mollura & Lungren, 2014; Mollura et al., 2013). However, these are the regions where they are most needed; with 94% of pregnancy and childbirth-related deaths occurring in developing countries and 80% of these occurring in areas of a high birthrate and limited access to healthcare (WHO et al., 2019). A major limitation of the availability of ultrasound is the lack of trained sonographers and healthcare workers (Maru et al., 2010; Darmstadt et al., 2009; WHO & UNICEF, 2018).

The acquisition of obstetric ultrasound requires a high level of training and expertise, with the sonographers iteratively repositioning the probe whilst viewing real-time at the surrounding anatomy to capture the most informative imaging plane. Additional difficulties include the movement of the fetal head and variable fetal position in the uterus. Thus, simplified scanning protocols have been published that do not rely on the intuition and experience of the sonographers, but where the probe is to scan along predefined linear sweeps across the body outlined solely by external body landmarks of the pregnant woman (Abu-Rustum & Ziade, 2017; Abuhamad et al., 2015; DeStigter et al., 2011). These scanning protocols (sweeps) allow for acquisition without highly trained sonographers, which can then be analysed by clinicians remotely with the advent of tele-medicine (Toscano et al., 2021; Marini et al., 2021), or by the medical image analysis algorithms.

In practice with these new acquisition protocols, we cannot assume the user will be able to differentiate between a usable scan and a low-quality non-informative one. Ultrasound scans are cheaply retaken during the same appointment, while retaking a scan after the patient has left is a large burden to the patient. Therefore, developing automated algorithms for ultrasound video quality assessment

is an important research challenge to aid in the adoption of ultrasounds where trained sonographers are not always available.

This paper seeks to provide a quality assessment method for sweep data. The proposed method utilises bounding box annotations of ultrasound sweep data, to train an anatomy detector model, and via kernel density estimation, quantifies how well the spatial and temporal position of bounding boxes fit in with the expected "typical" data.

## 2 RELATED WORK

Image quality assessment in signal processing is a well-researched topic with many different metrics proposed like PSNR (Peak Signal to Noise Ratio) and SSIM (Structural Similarity) (Wang et al., 2004). These however are image-based (not video), fully referenced methods (requires an undistorted reference image) and mostly focus on compression losses (Thung & Raveendran, 2009; Hore & Ziou, 2010).

No-referenced video quality assessment (NRVQA) literature presents models designed to quantify a specific type of image distortion such as blur (Marziliano et al., 2002), ringing (Feng & Allebach, 2006), blockiness (Wang et al., 2000) banding (Wang et al., 2016; Tu et al., 2020) or noise (Amer & Dubois, 2005). Current NRVQA metrics such as Li et al. (2019); Mittal et al. (2012) rely on using natural scene statistics and models of the human visual system. Thus they are specific for natural images, and likely will not generalise well to ultrasound videos, as these contain many types of distortions that are beyond the normal range for natural images.

Quality assessment of ultrasound video should therefore be defined differently, and should be based on clinical usefulness. This field has less research. Wu et al. (2017) and Lin et al. (2019) propose quality assessment of anatomy-specific criteria that can be separately assessed by individual networks; Wu et al. (2017) develop deep learning models to check things like if the fetal stomach bubble appears full and salient with a clear boundary, and Lin et al. (2019) "the lateral sulcus must be clearly visible" and "the skull is in the middle of the ultrasound plane and larger than 2/3 of overall fan shape area" and other criteria.

Zhao et al. (2022) pose the problem as an out-of-distribution detection task using a bi-directional encoder-decoder network, and using the reconstruction error as a quality metric. A larger reconstruction error means a lower quality. Saeed et al. (2021) used reinforcement learning to perform quality assessment of transrectal ultrasound images. The network is jointly optimised for a task predictor like classification or segmentation, whilst the agent learns which images will give the highest score for this task. Images giving a high score for the task are more likely to be higher in quality as the predictor is finding them easier to classify or segment.

The most related work is by Komatsu et al. (2021) who developed an object detector network to detect cardiac structures in fetal ultrasound scans and used the detection results to generate an "abnormality score" for the heart. It used a specific imaging plane of the heart, the three-vessel trachea view and the four-chamber view, where it expected to see a full set of specific anatomical substructures in every frame of these clips. Thus they generated an abnormality score between 0-1 (0 being normal) which decreased linearly with the number of anatomies and the number of frames detected. In our work, we do not expect to see a specified list of anatomy for a fixed number of frames, thus a score based on a simple linear formula cannot be used.

## 3 METHOD

This work uses bounding box annotations provided by an expert sonographer for a type of simplified ultrasound scan. The method first finds the distribution of the spatial and temporal position of the anatomies from the annotated bounding box data. It then evaluates how well the spatio-temporal position of the anatomies of a new video fits in the distribution.

### 3.1 DATA

The data was gathered as part of the Computer Assisted Low-cost Point of Care Ultrasound (CALO-PUS) Project (UK Research Ethics Committee 18/WS/0051). A simplified ultrasound scanning pro-

tocol proposed by Abuhamad et al. (2015) was refined by Self et al. (2022) to contain 5 steps which are shown in Fig. 1.

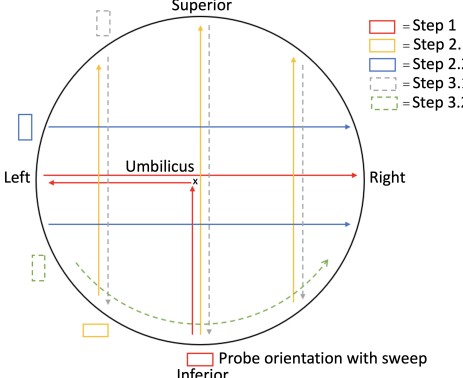 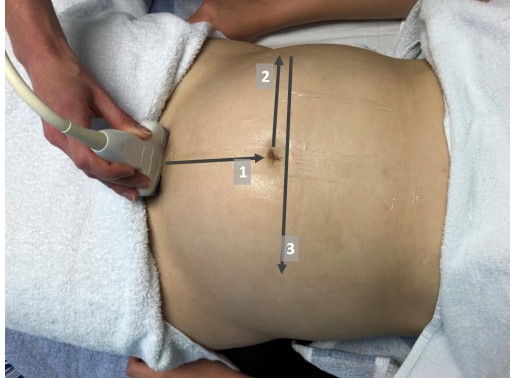

Figure 1: Left: all the different CALOPUS sweeps in Self et al. (2022) outlined in Section 3.1. Right: the T-shaped sweep (step 1 on the left) that we used in this work outlined on a participant.

The scans were taken by an experienced sonographer, who also annotated the scans frame by frame with a bounding box around each of the 11 possible anatomical structures (listed in Fig. 2). Of the five sweeps outlined in Fig. 1, this work only uses a T-shaped sweep (step 1). For this work, only scans of gestational age between 18-23 weeks, and cephalic presentation are used.

This was to ensure the scans looked as homogeneous as possible. With different fetal presentations, the imaging plane sweeps through the fetus at different angles, so we expect anatomies to show up at different locations on the screen and at different orders or timing throughout the scan. The gestational age cut-offs were chosen for similar reasons. This left us with 45 ultrasound videos, each around 40 seconds long.

Whilst we use only sweep 1 (see Fig. 1), this work could easily extend to any other sweeps. We chose the T-shaped sweep because it often showed the most anatomies, thus bounding boxes. The different bounding box annotations consisted of 11 anatomies, of which were distributed as shown in Fig. 2.

The CALOPUS project contains a further 72 annotated scans from India (from THSTI (Self et al., 2022)) that fit the gestational age and fetal presentation criteria. The video content in these scans cannot be viewed, however, the annotation information can be used: the bounding box coordinates, anatomy label and frame number.

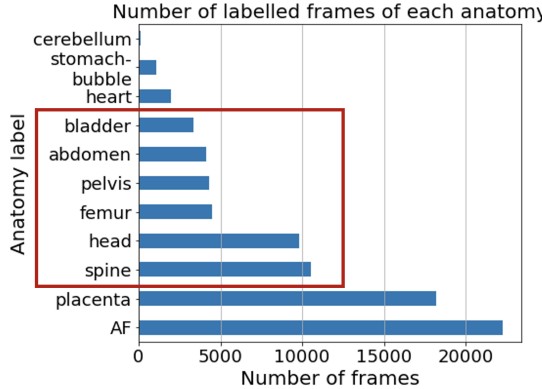

Figure 2: Number of frames per anatomy in the dataset after applying exclusion criteria in Section 3.1.

## 3.2 OVERVIEW

A brief overview of our method is: (1) using the bounding box annotated videos, we train an object detector that can produce bounding boxes around fetal anatomies. (2) Estimate the probability distribution of spatial position and timing of the bounding boxes for a cephalic T-shaped sweep using kernel density estimation. (3) Perform inference on new data with our trained anatomy detector model to get bounding boxes for each anatomy for this video. (4) Compare the spatial and temporal position of the bounding boxes of the new video against the distribution we estimated in (2). If the new bounding box properties fit in well with our estimated distribution, we propose it is of high quality, whilst if it doesn't, we propose it is abnormal or low quality. Numerical values for 'how well it fits the distribution' can be produced via calculating the probability of the bounding box having a

property of a certain value or a less likely value (p-value) via integration of the probability density function (PDF). This gives one value for each bounding box and averaging the probability over the entire video clip gets our quality metric.

### 3.3 ANATOMY DETECTION MODEL

Our anatomy detection model is based on the RetinaNet architecture with focal loss (Lin et al., 2017). This is a one-shot detector, that contains feature pyramid networks to propagate multiple scale features down the network. The network has subnets for object classification and bounding box location regression. The backbone used was simply a ResNet-50 architecture (He et al., 2016) that was pre-trained on the ImageNet1k image classification dataset. The overall network was pre-trained with the COCO dataset to achieve a mean average precision of the bounding box of 36.9 in the COCO dataset. This is a one-shot detector and can do real-time detection. Wu et al. (2019)'s Detectron2 code was used for this.

The data imbalance as seen in Fig. 2 caused large drops in performance; anatomies such as the cerebellum and stomach bubble were never detected, whilst the placenta and AF were over-predicted. Thus, we use a subset of only 6 anatomies: the spine, head, femur, pelvis, abdomen and bladder. There is still a considerable data imbalance, but not of the scale as with all 11. These anatomies were chosen as they were consistently showing in the majority of the scans, and were not within each other, i.e., the stomach bubble will be within the abdomen and cerebellum within the head.

### 3.4 DEFINING BOUNDING BOX PROPERTIES

We chose to use the temporal and spatial position of the bounding boxes as discriminators to determine if a scan is 'typical' or not. There are other properties that could have been used: size, aspect ratio of the boxes, as well as properties of the image content inside the bounding box like texture and pixel intensity. The bounding box annotations are not tightly aligned to the exact edge of the anatomy structure, but often loosely drawn to the approximate location of the anatomy, so do not contain the precise size, and aspect ratio information of the anatomy. Image-based features are not used because we want to only use information from the bounding boxes (not the image)lso we could leverage the extra data from THSTI, and keep the method simple and efficient.

### 3.5 PROBABILITY DENSITY FUNCTION OF BOUNDING BOX PROPERTIES

We use kernel density estimation (KDE) with a Gaussian kernel to estimate the probability distribution function (PDF) of the timing and position of the bounding boxes. A plot of the PDF and histogram for the timing of the femur is shown on the left of Fig 3.

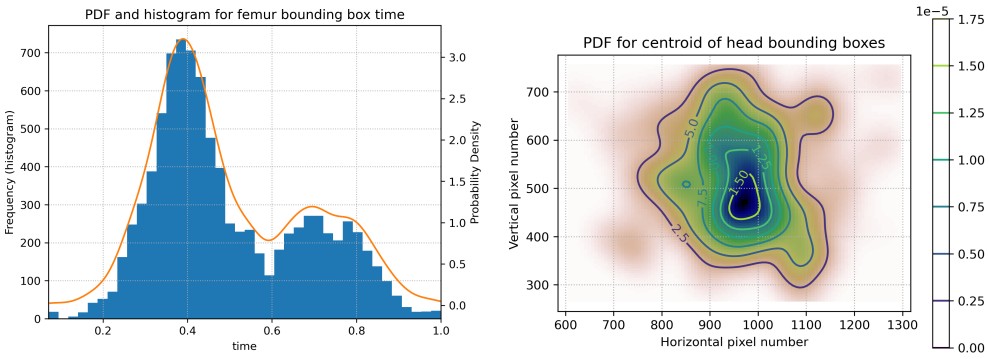

Figure 3: Left: histogram and PDF of the timing for the femur. Right: the joint PDF for x, y coordinates of the head bounding boxes.

However, we cannot assume these properties of bounding boxes are independent, rather, it is likely they are highly dependent. For example, due to the shape of the sweep, the timing and the position of the bounding boxes interact; as we pan side to side (the top of the "T" of the sweep), the anatomies first go right, then left in that order, thus the x co-ordinate and the time of the bounding boxes are

highly correlated. We account for these dependencies by, instead of having a PDF for each property, having a joint multi-dimensional PDF for all properties. A two-dimensional PDF of the x and y coordinate of the head bounding boxes is shown on the right of Fig. 3

We extend this to a three-dimensional joint PDF, with the axes as: x, y-co-ordinate and time. This is a three-dimensional function, so cannot be visualised easily, but plotting this on top of a frame at different time points gives Fig. 4.

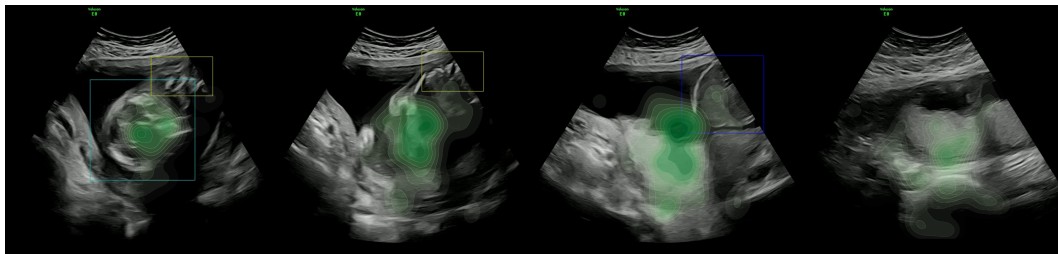

Figure 4: The change across time for the contour of the PDF for the spatial position of the head in the context of the actual ultrasound frame. From left to right, the time point in the scan increases. The stronger the green colour, the higher PDF values. The blue and yellow boxes are head and spine bounding boxes respectively.

The bandwidth of the kernel used in the KDE strongly influences the overall PDF - much more so than the shape of the kernel (Turlach et al., 1993). In this work, we use a Gaussian kernel and fine-tuned the bandwidth using contextual knowledge. Both Silverman's rule and Scott's rule gave very similar PDFs, however the PDF was very peaky and multi-modal as shown in the rightmost image in Fig. 5, which shows steep decreases in the PDF value within a 20-pixel radius of the mode. We believe it is wrong to think a spine that appears 20 pixels away from this peak indicates a much lower-quality scan, we don't think sweep scans are that fine-grained. We believe there are two general regions within the fan-shaped area of the ultrasound where we expect to find a spine during the sweep, so we adjust the bandwidth to reflect this. With too large bandwidth, we lose spatial resolution. Similar intuition was used for the temporal domain too.

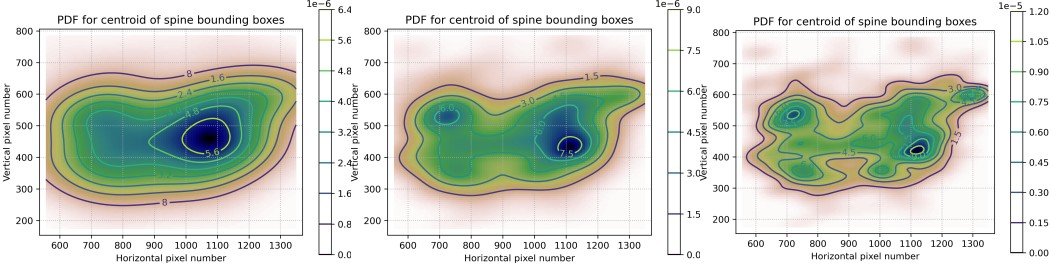

Figure 5: Contour plots of the PDF for the spatial position of the spine bounding boxes. The PDF becomes less peaky as the kernel bandwidth decreases from left to right. The right PDF was produced when using Scott's Rule for bandwidth selection.

### 3.6 INTEGRATION OF THE PROBABILITY DENSITY FUNCTION

To get a quantitative value of how well the position and timing of the bounding box fit in with the estimated distribution, we cannot simply use the probability density value of the bounding box because the value is determined by the PDF integrating to 1, so the values cannot be compared for different PDF shapes [1]. Instead, we can use the probability of the bounding box having this position and timing or a less likely timing/position (equivalent to a p-value) to get a number from $0 - 1$. Thus we integrate the PDF for all areas where the PDF evaluates to a lower probability density than that

---

[1]The probability density/peak PDF value could be used, but this skews most of the values to be very low - as shown in appendix Fig. 9

of a given position and timing, i.e.:

$$p = \iiint_V f(x, y, t)\, dx\, dy\, dt \tag{1}$$

where $f$ is the PDF, $x, y, t$ are the $x$ and $y$ coordinates, and time of the bounding box respectively, and the limits of integration $V$ is the volume/region in $x, y, t$ domain that encloses:

$$V = V(x, y, t) \quad \text{where:} \quad f(x, y, t) < f(x_{bbox}, y_{bbox}, t_{bbox}). \tag{2}$$

Rather than direct integration of the PDF to find the probability, we used Monte Carlo sampling methods to estimate the probability that the integral evaluates to. The corresponding algorithm is written in pseudo-code in the Appendix.

This method relies on using many random samples of the PDF to get an estimate of the PDF, thus becoming more accurate and precise with more samples. Although sampling many times takes time, it takes much less time than directly integrating via built-in integration libraries as we don't require such exact solutions. With this method, we can change the precision with the number of samples we use. As we require one integration with each bounding box, and there often is multiple bounding boxes per frame for each video that is around forty seconds long, we compromise between run time and precision. A plot of precision and number of samples for the same integral is shown in Fig. 6.

Whilst the time taken for each integration is directly proportional to the sample size, the error is inversely proportional to the square root of the sample size, thus we compromised with 1,000 samples for each integral.

To get from probabilities to a quality score, we simply took the mean probability for video across all the anatomies. Therefore, our quality score is a measure of how similar the bounding boxes of the new scan are to our annotated scans. All our annotated scans were screened before annotation, where non-typical, unusual scans were discarded, to ensure the quality. Even with some non-typical scans present, if most scans were ideal, with enough data the resulting PDF from KDE should not be affected much by the few non-typical scans in the annotated data.

Our quality metric uses only bounding box properties without explicitly requiring manual annotations of the quality score by an expert. Additionally, basing a quality score off the bounding boxes makes sense as these bounding boxes contain the clinically important features in the video, and the rest outside the bounding boxes are not focused on by the sonographer.

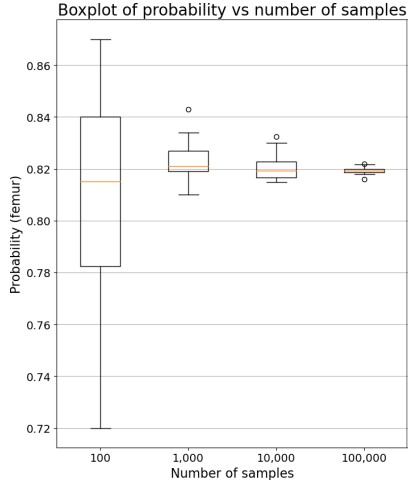

Figure 6: Box plot of an arbitrary integral (an arbitrary femur position and time) performed 30 times, but with increasing sample number for the Monte-Carlo method.

## 4 EXPERIMENTS AND RESULTS

### 4.1 ANATOMY DETECTION MODEL

To train the anatomy detector model, the data was split at the patient level, with 31:7:7 patients respectively for training, validation and testing. Simple data augmentation was used, including random horizontal flips, brightness, crops, and slight rotations. The minority classes were not over-sampled or over-augmented, but proportionally sampled. The network was trained for 80 epochs, with an initial learning rate of 0.001, which dropped to 0.0001 at 40 epochs. The batch size is 16, and the momentum is 0.8. Early stopping was used, so the best validation model was saved. The intersection-over-union results of the trained model are shown in Fig. 10 and confusion matrix in Fig. 11 in the Appendix.

We can assess how well the detector performs for our purpose by comparing it with a "perfect" detector (the ground truth annotations). A scatter plot of the quality probability score for each

bounding box against time can be used for this comparison. If the plots look similar and mean probability values are similar then the detector is good enough. The comparison is shown in Fig. 7 for one of the ultrasound videos.

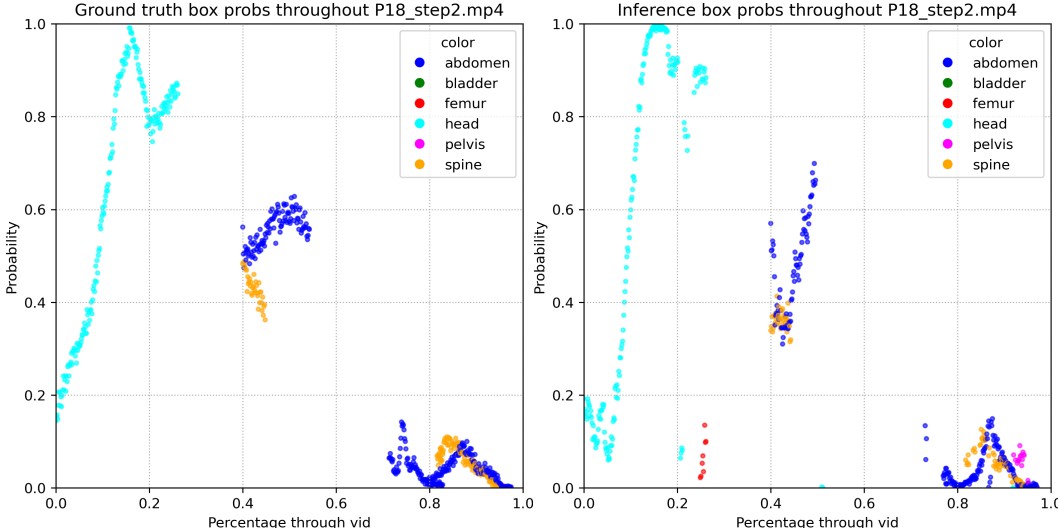

Figure 7: The vertical axes shows how well the spatial and temporal position of the anatomy fits with our seen data. The higher up on the vertical axes, the better the fit with our distribution from the PDF. Each point in the plot is a bounding box of an anatomy.

We compare the mean and standard deviation of the probabilities between the ground truth and our trained model in Table 1. We also include a trained Faster R-CNN model (Ren et al., 2015) to view the robustness of the methods to the detector model used. Lastly, we include a very under-trained RetinaNet model with less than 10% the training time of our final model used, to view as a baseline, and see the effect of our fine-tuning. A model with no fine-tuning and that has only been pre-trained on natural images did not produce any bounding boxes at all for our ultrasound videos.

Table 1: Table of the mean and standard deviation of the probability score for our test set videos for the various models vs ground truth. Our final model used was the RetinaNet.

| Video | Mean Probability | | | | Standard Deviation of Probability | | | |
|---|---|---|---|---|---|---|---|---|
| | GT | RetinaNet | Faster R-CNN | Undertrained | GT | RetinaNet | Faster R-CNN | Undertrained |
| P15 | 0.213 | 0.232 | 0.283 | 0.181 | 0.192 | 0.216 | 0.247 | 0.246 |
| P18 | 0.322 | 0.301 | 0.274 | 0.103 | 0.313 | 0.331 | 0.299 | 0.159 |
| P54 | 0.167 | 0.251 | 0.254 | 0.128 | 0.276 | 0.294 | 0.290 | 0.167 |
| P88 | 0.246 | 0.238 | 0.156 | 0.087 | 0.183 | 0.207 | 0.148 | 0.084 |
| P115 | 0.308 | 0.372 | 0.312 | 0.154 | 0.298 | 0.297 | 0.281 | 0.253 |
| P144 | 0.453 | 0.450 | 0.391 | 0.256 | 0.279 | 0.236 | 0.256 | 0.345 |
| P163 | 0.406 | 0.274 | 0.308 | 0.115 | 0.310 | 0.196 | 0.238 | 0.218 |
| P166 | 0.466 | 0.515 | 0.415 | 0.215 | 0.284 | 0.279 | 0.308 | 0.107 |
| Avg(x - GT) | 0 | 0.0065 | -0.0235 | -0.1678 | 0 | -0.0099 | -0.0085 | -0.0695 |

## 4.2 PROBABILITY MODEL AS A METRIC FOR QUALITY ASSESSMENT

In this work, no videos have been annotated with an explicit quality score since they are all screened as part of the data-gathering process. Therefore we use other types of sweeps (see Fig. 1) and other fetal presentations as our "bad quality" scans, for evaluation of our method. The anatomy of these different scans should appear at different locations and timings and our method should differentiate between these.

We run our method on 7 breech presentation step 1 videos and 3 videos each of step 2.1, 2.2, 3.1 (Fig. 1), all in cephalic presentation. Thus, in these results, the mean probability should be

noticeably higher in step 1 cephalic than any others (similar to one class classification). A box plot of the probability scores is shown on the left of Fig. 8.

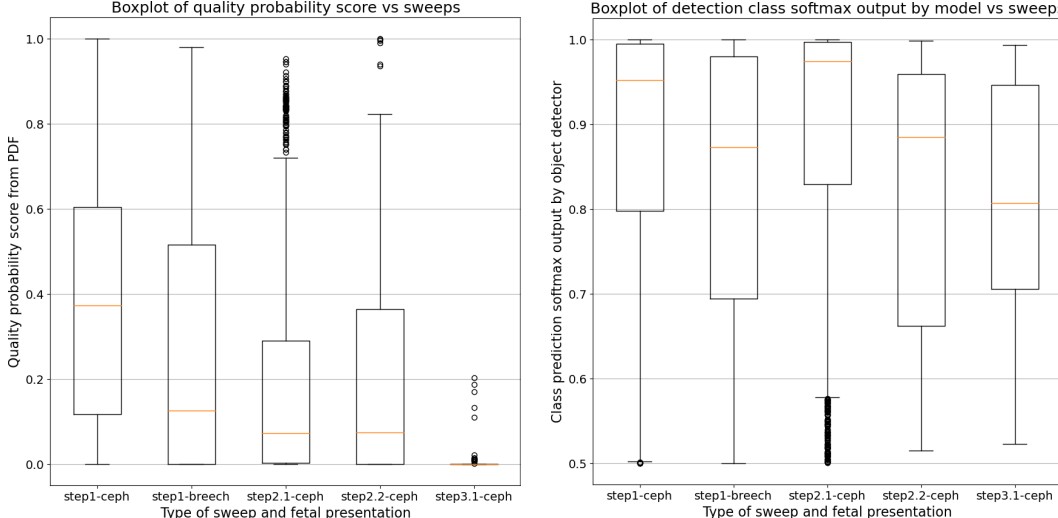

Figure 8: The left is a box plot using the ground truth bounding box annotations for the videos. And the right shows the mean detection class softmax outputs from the detection model. Step-1-ceph means step 1 in Fig. 1 with a cephalic presentation fetus.

To have a comparison we use the softmax output by the detection model for the class of the object. From intuition, if the detector sees a head that is very similar to all the heads it has been trained on, then the model will confidently predict that structure as a head, and so the softmax output for the head class for this bounding box will be almost 1. However, less familiar-looking heads will have much lower values. With different types of sweeps and different fetal presentations, the ultrasound view slices the anatomy differently, so the anatomies look different. Thus we expect lower class confidence for these other sweeps and presentations.

The right plot in Fig. 8 shows that there is no obvious difference between the mean values of step 1 cephalic sweeps vs any of the others using this class confidence method, but our method (left) shows a visible difference. We perform a Welch's T-test where we treat each video mean probability as a sample, and use a one-tailed alternative hypothesis (step-1 cephalic is higher), which finds that the p-value is 0.0341, thus there is a significant difference in the quality score between the step 1 cephalic sweeps and the rest. With the confidence method, we find the p-value of 0.396 using the same T-test which isn't enough evidence to say that the means are significantly different.

## 5 CONCLUSION

We present an ultrasound video quality assessment method, by using kernel density estimation to assess whether the spatial and temporal positions of anatomies in a specific scan follow the typical distribution. We compare this to the approach using the detector class softmax value and we show our method is effective at discriminating between different sweep steps and fetal presentations. We hope that this could be used not only for detecting different types of sweeps, but detecting non-typical, unusual scans from the same sweep as an automated quality screening process to ultimately aid in the adoption of sweep ultrasound in regions where there is a lack of trained sonographers.

ACKNOWLEDGMENTS

We thank the reviewers for their helpful feedback. Jong Kwon is supported by the EPSRC Center for Doctoral Training in Health Data Science (EP/S02428X/1). CALOPUS is supported by EPSRC GCRF grant (EP/R013853/1).

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

# A APPENDIX

## A.1 MONTE-CARLO SAMPLING

Pseudo-code of the Monte-Carlo sampling used to estimate the probability

```
# We want to find the area under the pdf
for all x,y,s,t where the pdf evaluates to
a value lower than our bounding box p_density.

function pdf(x,y,t) #input shape:  (3,)
return prob_density   #output shape: (1,)

# is a function returns the probability
density for the pdf evaluated at (x,y,t).

# Evaluate the pdf for new bounding box
x,y,t:
p_density_of_bbox = pdf(x_bbox,y_bbox,t_bbox)

# Randomly sample from your pdf 10000 times
to get a list of x,y,t data:
samples_xyt = sample(pdf, size=10000)    # shape = (3,10000)

# The p of the sample evaluating to below
our p_density_bbox = value of the integral.

# So we count the number of samples where
p_density_sample < p_density_bbox and divide
by the number of all the samples, to get an
estimate of this probability:
lowsamples = pdf(samples_xyst) < p_density_of_bbox
# lowsamples is a 10000 logical vector
integral = sum(lowsamples) / 10000
```

## A.2 PROBABILITY DENSITY VS USING A P-VALUE

Fig. 9 shows a comparison of using the p-value vs using the probability density / max probability density. As one can see from the graphs, the right side is a scaled down version of the left side, however the amount of scaling is non-linear and depends on the the shape of the PDF. The full scale from 0-1 is rarely seen on the right side. Using the left side seems appropriate.

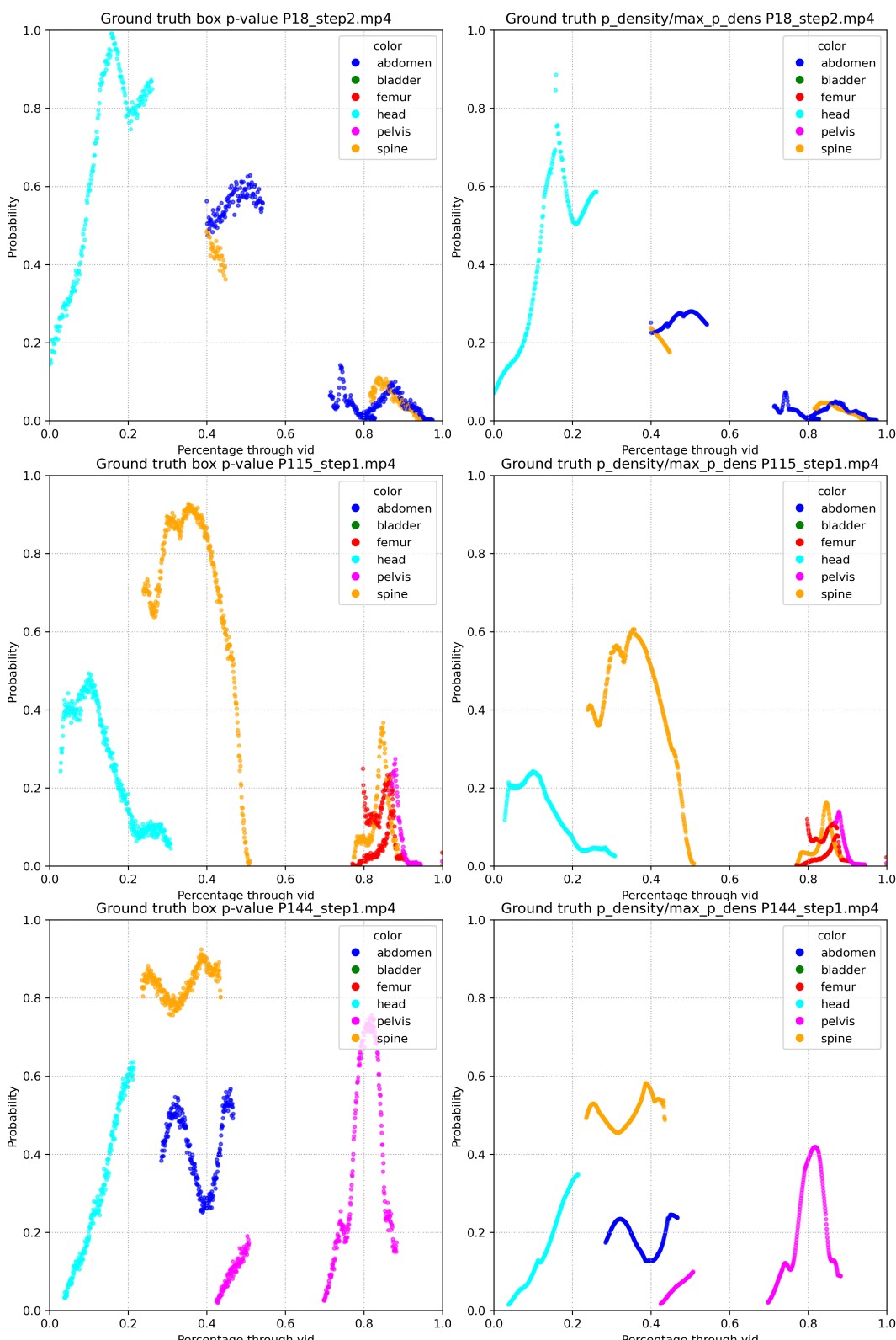

Figure 9: The right shows how the ground truth box p-values vary. This is calculated via estimating the integral via Monte-Carlo methods outlined in Section 3.6, whilst the left shows the probability density of the bounding box / max probability density.

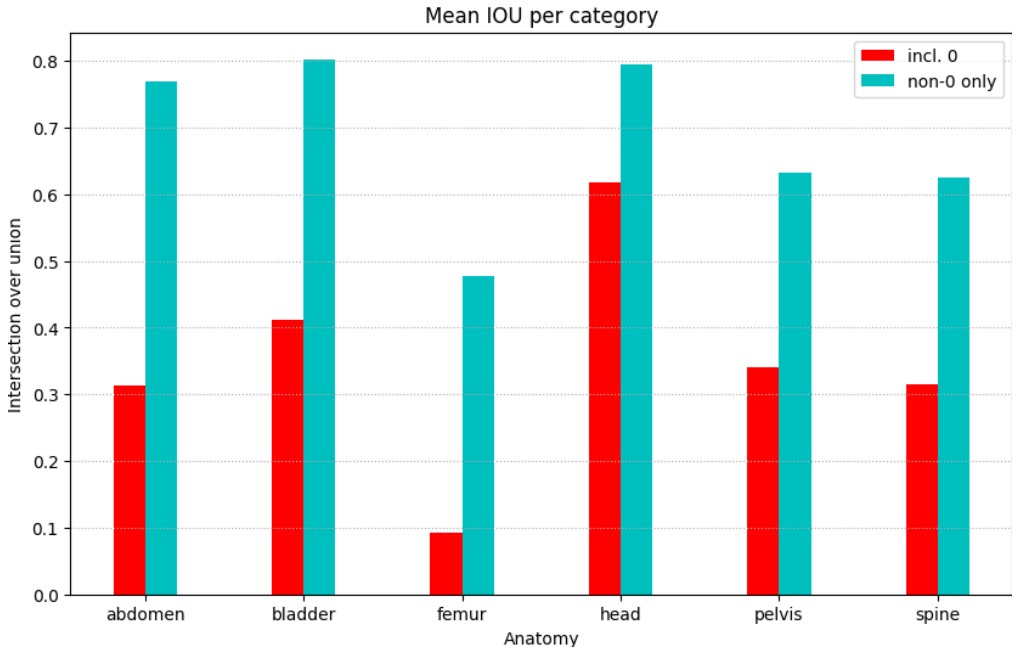

Figure 10: The red bars show the intersection-over-union of the bounding boxes for each anatomy where the no prediction of a box counts as 0 intersection. The cyan bars show only the intersection-over-union of the boxes that have been predicted.

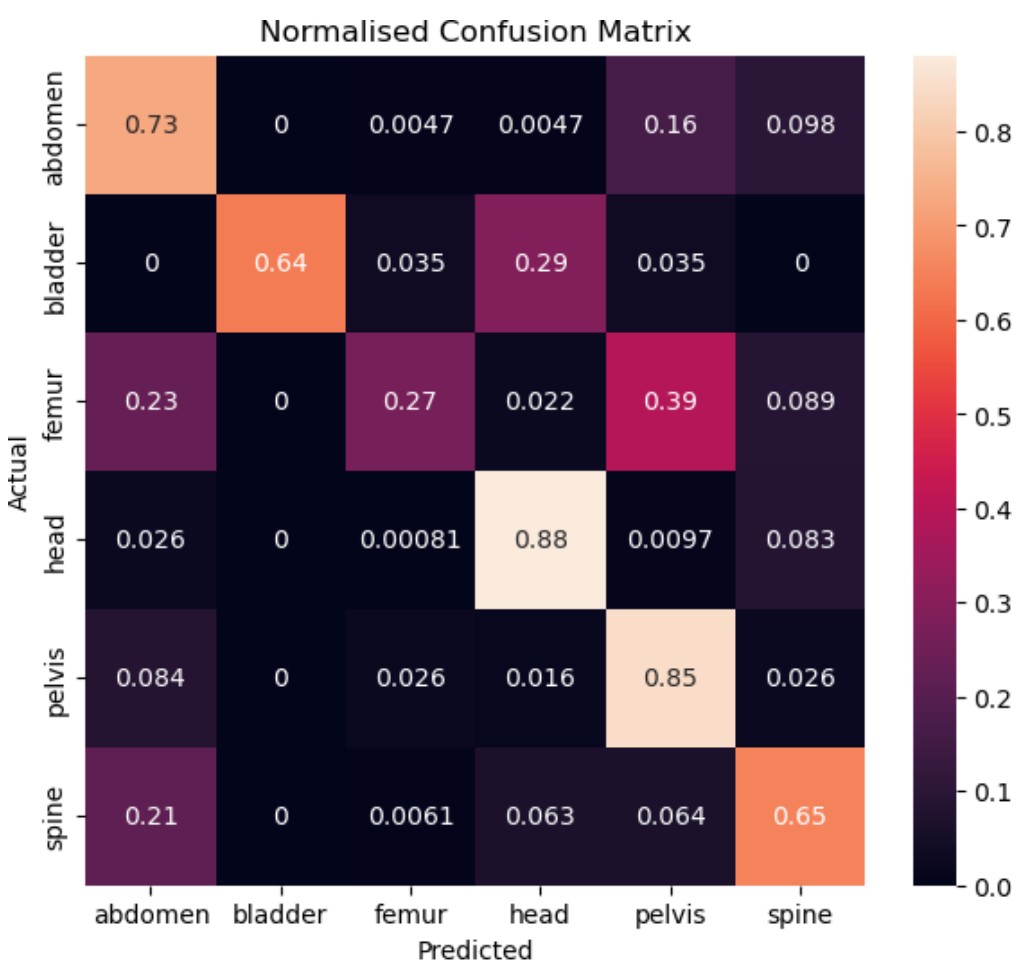

Figure 11: Confusion matrix of the object detector model.

