# OpenReview forum: "A Kernel Density Estimation based Quality Metric for Quality Assessment of Obstetric Ultrasound Video"
_ICLR.cc/2023/Workshop/TML4H — ICLR 2023 Workshop TML4H Poster_

### Official Review · Reviewer_6Nau · 2023-03-03
**An interesting topic, generally well-organized but needs more conclusive evidence/results.**

**Rating:** 6
**Confidence:** 3

**Review:**

The paper presents an automated quality assessment algorithm to evaluate the usability of ultrasound scanning videos. The authors developed spatio-temporal Gaussian PDFs to model the properties of the anatomy bounding boxes, based on which they propose to quantify the 'quality' of a scan with an integration scheme.

Strength:
The idea is interesting and persuasive, which aims to evaluate the scan in a real-time manner to avoid reworking. The evaluation model is designed in a simple yet effective way, and the authors clearly illustrate how they devise the evaluating scheme against clinical challenges. The paper is generally well-organized.

Weakness:
1. There are several typos/confusions that need to be revised.
   Figure 2's caption remains some irrelevant information.
   The title of Figure 7's left-side sub-figure is the same as the right-side one. It's hard to tell the result from detector predictions and the one from ground-truth results. Please fix this.
2. More details about the initialization of PDF are needed.
3. Authors should explain why the evaluation function is based on the integration of 'less-likely samples' rather than the primitive probability given by PDF itself. The necessity of using the 'p-value' measurement needs to be illustrated. Also, I suggest the authors investigate the comparison between the integration and the PDF's probability.
4. The results of Table 1. are not sufficient enough to support how well the detector performs. For example, the author can consider investigating a randomly initialized detector to see the baseline performance so that the effectiveness of the detector can be compared.
5. The paper did not give a conclusive suggestion on how to apply the metric for quality. For example, is there any threshold that can be considered to mark the 'good' scan?
6. The sub-figures in Figure. 8 lack statistical significance analysis. For example, T-test or Wilcoxon test.
7. Is the quality probability an integration of all types of anatomies? The 'mean probability for the video' is vague as there are temporal changes and cross-anatomy variables.

---

### Official Review · Reviewer_vokd · 2023-03-04
**Review of Paper22**

**Rating:** 6
**Confidence:** 4

**Review:**

**Summary of The Paper**:

The paper proposes a method that utilizes a kernel density estimation based quality metric to identify high-quality obstetric ultrasound videos. Specifically, a detection model is first applied to extract bounding boxes of different anatomies, and then, based on the estimated distribution, the probability score of each anatomy is integrated across the whole video.

**Strengths**:

1. The paper proposes a method to use the kernel density estimation based approach to assess sweep data quality and results in good performances.

2. The paper provides detailed descriptions of data acquisition and the experimental setup.

3. The paper also includes some visualizations of the PDF, which facilitate understanding of the method.

**Weaknesses**:

1. There are some typos and confusing points in the paper, such as 1) in the caption of Figure 2, the last sentence appears to be a revision comment that hasn't been deleted; 2) in Figure 2, the red box does not include 'head'; 3) in Figure 4, the blue and yellow boxes' representation is not clear; 4) in Figure 8, it is mentioned that "the right shows the quality score using the model's detected bounding box," but this has not been included in the figure.

2. For other types of sweeps, it is unclear if the detector model is retrained for each type or if the previous model trained on high-quality data is used to extract the bounding box. If the model is not retrained, it is possible to result in bad performances, thus being the reason for the low quality scores for other different types.

3. As mentioned in the paper, "we expect the step 1 cephalic sweeps to have the highest mean probability score." Does this imply that if we want to find a high-quality video, we only need to identify if it is scanned by step 1 cephalic sweeps? Can this be achieved simply by a classification model?

4. In Table 1, the results of most of the test samples align well with the ground truth except for the video P163_step1. It is unclear whether the detection failed or if there are other reasons for the discrepancy. The authors should provide more comments on Table 1. Additionally, it would be helpful to show explicit evaluations of detection performances (e.g., mAP) for each sample.

**Other Comments/Questions**:
1. Are all the videos of the same length or have the same number of frames?

2. Is there a preferred age range for patients to be included in the experiments?

3. Will this algorithm work if the fetuses are twins?

4. In Figure 8, is the quality probability score an average of different anatomies in the corresponding type? If so, are there any differences among different anatomies?

---

### Meta-Review · Area_Chair_2jUY · 2023-03-05

**Recommendation:** Accept (Poster)
**Confidence:** 5

**Metareview:**

This paper proposes a new quality metric for assessing the quality of obstetric ultrasound videos. In general, this paper is well-received: both reviewers appreciate its simplicity and effectiveness, and rate it as marginally above the acceptance threshold,

There are some minor concerns about typos and confusing points. The authors should carefully address them in the final version to improve the paper's quality.